# Ketone displacement and migration enabled by trifunctionalization of vinyl triflates

Shiyang Wang[1,2], Tong Yao[1,2], Yu Liu ⬡[1] ✉, Yangyang Li ⬡[2] ✉ & Guoyin Yin ⬡[2] ✉

Functional group displacement and migration represent powerful, yet underexplored strategies in synthetic chemistry, offering unique opportunities for molecular diversification and drug discovery. Here, we report a nickel-catalyzed deoxygenative trifunctionalization of vinyl triflates, which enables the efficient synthesis of structurally diverse, boron-containing poly-substituted cyclohexanes featuring quaternary carbon centers. This reaction is a key step enabling aryl displacement of a ketone group and its migration to an adjacent carbon center. Notably, the transformation exhibits broad substrate scope and exceptional, programmable diastereoselectivity in arylative ketone migration. Moreover, this transformation enables efficient $\alpha$-arylation of unsymmetrical ketones with excellent regio- and diastereoselective control—an outcome that remains challenging to achieve using existing methods. Furthermore, this strategy is particularly well-suited for the late-stage functionalization of structurally complex bioactive molecules, facilitating the rapid generation of analogs.

The displacement and migration of key functional groups within complex bioactive molecules have emerged as powerful strategies in contemporary drug design and discovery, offering the potential to significantly enhance or even transform biological activity (Fig. 1a)[1–9]. For instance, substitution of a C1–NH group with a methyl group followed by migration to the C2 position resulted in a 365-fold increase in CCR6 antagonist potency[10]. Likewise, replacing a C1–O group with a benzene ring and relocating it to C2 produced a twofold improvement in androgen receptor modulation[11]. Notably, exchanging a ketone with an alkene moiety, followed by positional migration, converted monoamine oxidase inhibitory activity into selective cyclooxygenase-2 inhibition (Fig. 1b)[12,13]. Consequently, developing a simple and versatile procedure that facilitate the selective displacement and migration of key functional group within complex molecular architectures is highly valuable.

Ketone groups are among the most prevalent and synthetically valuable functionalities found in pharmaceuticals and bioactive molecules[14–16]. The ability to precisely manipulate these moieties, particularly through 1,2-ketone migration, has emerged as a powerful strategy for late-stage functionalization[17,18]. For example, the Dong group reported a Catellani-type process that enables 1,2-carbonyl transposition with enhanced modularity and efficiency[19]. Later, the Bhawal and Morandi group developed a complementary approach grounded in Willgerodt–Kindler reaction, providing an alternative route for ketone migration[20]. Although these pioneering strategies laid important groundwork, achieving general displacement and migration of ketone groups remains underdeveloped[21,22]. On the other hand, vinyl triflates have been readily prepared from ketone precursors and, as pseudohalides, have participated in metal-catalyzed cross-coupling reactions via C–O bond activation, enabling the efficient construction of various C–C and C–X bonds[23–25]. However, to date, the trifunctionalization of vinyl triflates has remained rarely explored and continues to represent a significant challenge. In line with our ongoing interest in nickel-catalyzed alkene difunctionalization[26–29], we have developed a nickel-catalyzed deoxygenative trifunctionalization of vinyl triflates that allows the synthesis of boron-containing polysubstituted cyclohexanes featuring quaternary carbon centers[30,31] under mild conditions (Fig. 1c). Importantly, this transformation provides a strategic platform for programmable arylative ketone migration, proceeding with excellent regio- and stereoselectivity.

[1]College of Chemistry and Life Science, Advanced Institute of Materials Science, Changchun University of Technology, Changchun, Jilin, China. [2]The State Key Laboratory of Metabolism and Regulation in Complex Organisms, The Institute for Advanced Studies, Wuhan University, Wuhan, Hubei, China. ✉e-mail: yuliu@ccut.edu.cn; yangyangl@whu.edu.cn; yinguoyin@whu.edu.cn

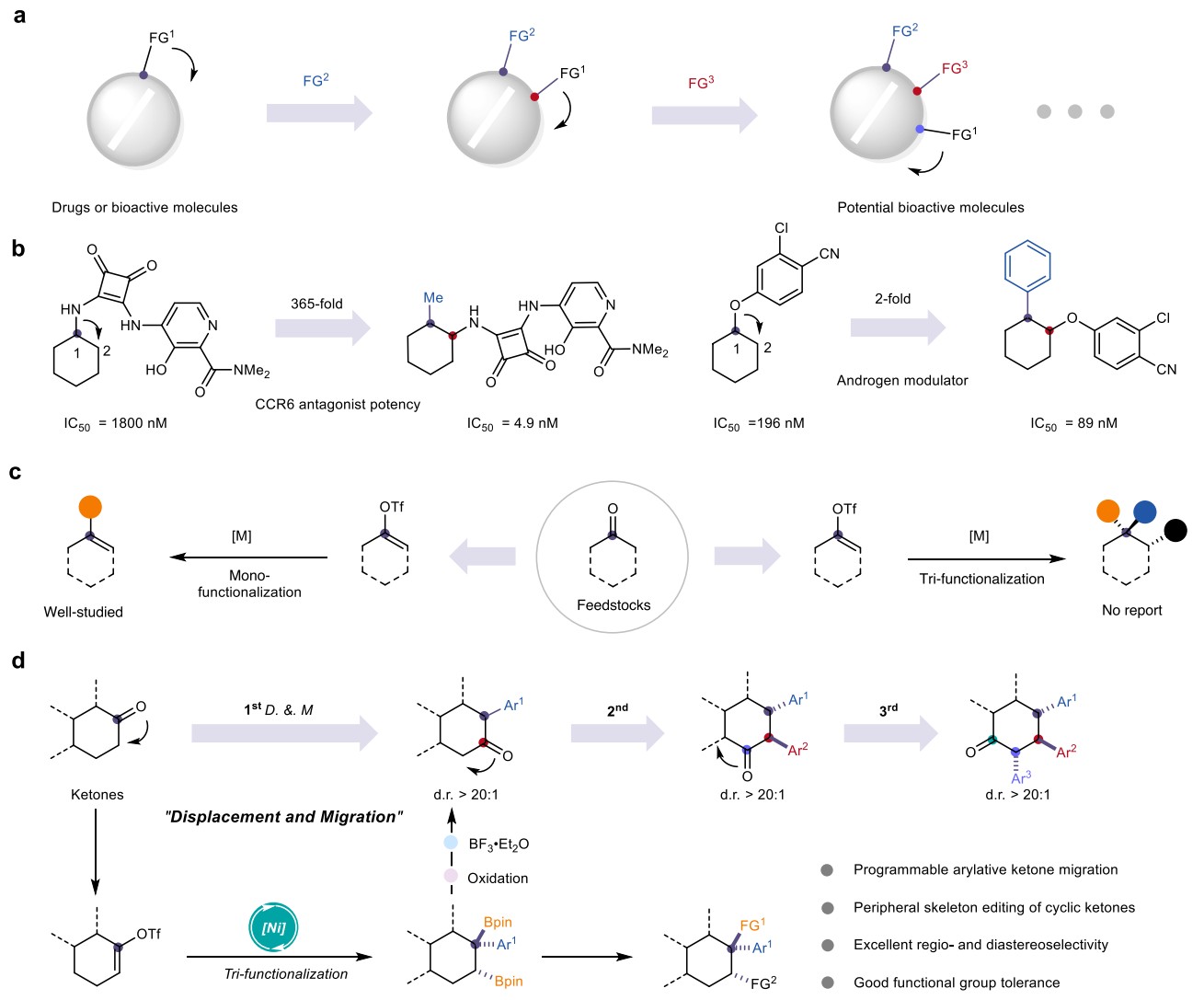

**Fig. 1 | Systematic modification via functional group displacement and migration. a** Functional group consecutive displacement and migration. **b** Representative bioactive molecules involving functional group migration. **c** State-of-art of vinyl triflates functionalization. **d** Programmable arylative ketone migration based on a developed Ni-catalyzed deoxygenative trifunctionalization of vinyl triflates (this work).

## Results

### Reaction development

Given the ease of accessing vinyl triflates from ketone precursors[23,24,32,33], our investigation began with the use of commercially available vinyl triflate (**1a**), bis(pinacolato)diboron (B$_2$pin$_2$, **2**), and bromobenzene (**3a**) as model substrates. After extensive optimization, we identified the combination of NiCl$_2$·DME as the catalyst, LiOMe as the base, and $^s$BuOH as the solvent at 60 °C as the optimal reaction conditions. Under these conditions, the desired trifunctionalized arylboration product (**4a**) was obtained in 82% yield with excellent diastereoselectivity. As illustrated in Fig. 2, replacing NiCl$_2$·DME with other nickel sources led to diminished yields. When a palladium catalyst was employed, only the boron–Heck-type byproduct was observed. Solvent screening revealed that alcohols are crucial for the reaction's efficiency, with $^s$BuOH proving to be optimal. Notably, the reaction also proceeded well under ambient air or in the presence of stoichiometric water, with only a slight decrease in yield, highlighting the operational robustness and practicality of the protocol.

### Mechanistic investigations

To gain mechanistic insight into this transformation, a time-course study was performed using model substrates (Fig. 2b). Vinyl triflate **1a** was

rapidly consumed, accompanied by a steady increase in the intermediate vinyl boronate **4aa**, which peaked and then gradually declined over nine hours. Concurrently, the desired product **4a** accumulated continuously throughout the reaction. This result suggests that vinyl boronates likely serve as key reaction intermediates in the reaction pathway. To validate this hypothesis, vinyl boronate **4aa** was subjected to the standard catalytic conditions, affording the desired product **4a** in excellent yield, consistent with its role as a reaction intermediate. Based on these results, together with precedents in nickel catalysis[34–39], we proposed a plausible catalytic cycle for this reaction (Fig. 2c). The cycle is initiated by transmetallation of B$_2$pin$_2$ with the nickel catalyst (**I**) to generate a [Ni]–Bpin species (**II**). Subsequent the vinyl boronate intermediate **4aa** was formed through deoxygenative borylation. This intermediate then undergoes a second migratory insertion into [Ni]–Bpin to form species **III**, which reacts with **3a** to deliver the target product **4a**, while regenerating the active nickel catalyst (**I**).

### Reaction scope

With optimal reaction conditions established, we first investigated the substrate scope of aryl bromides in this catalytic system, all of which exhibited excellent regio- and diastereoselectivity (>20:1 r.r. and d.r.). As shown in Fig. 3, aryl bromides bearing various functional groups,

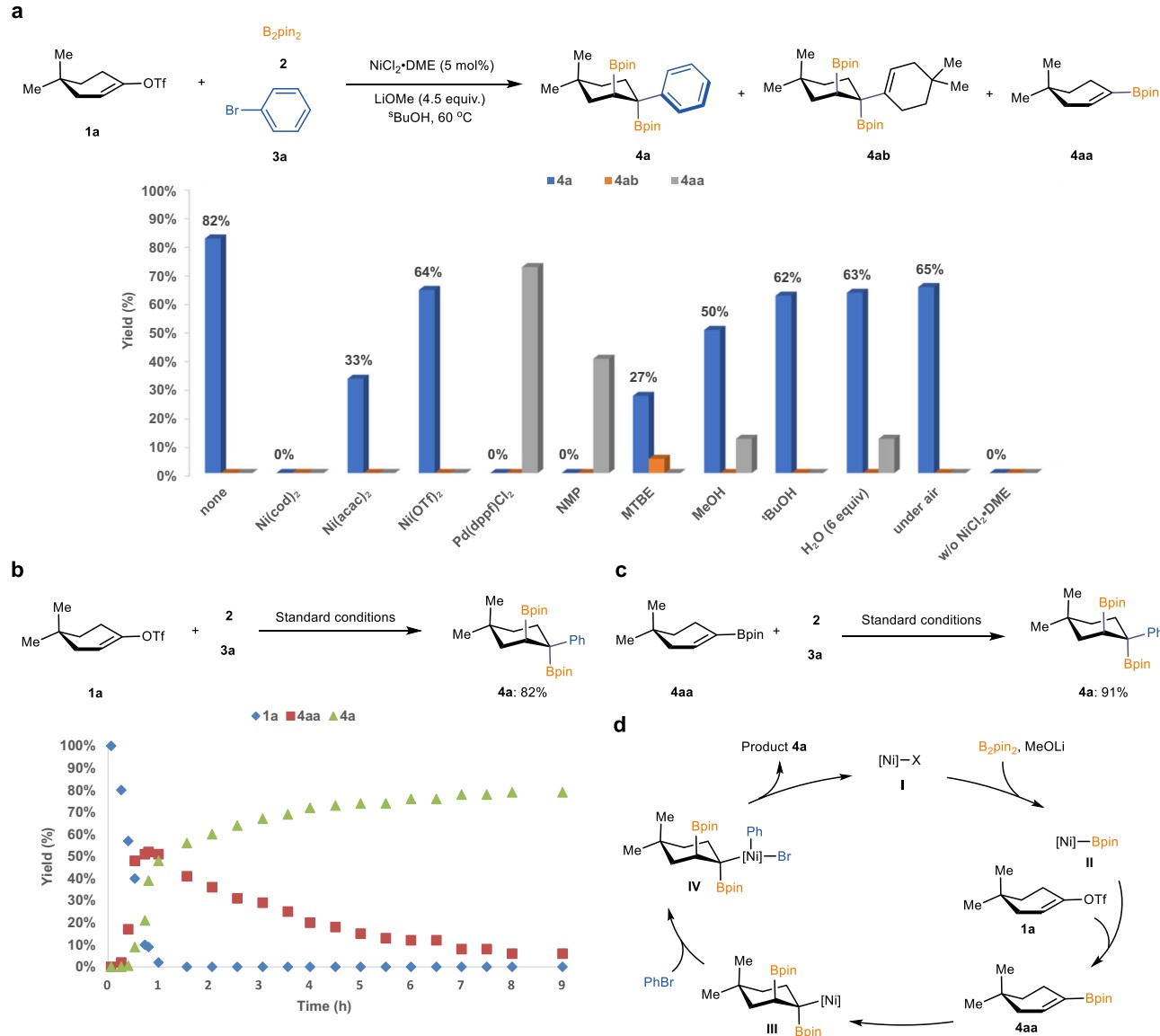

**Fig. 2 | Reaction development. a** Reaction optimization. **b** Study of time course. **c** Reaction with vinyl boronate intermediate. **d** Proposed catalytic cycle. General reaction conditions: NiCl₂·DME (5.0 mol%), **1a** (0.4 mmol, 1.0 equiv.), **2** (1.6 mmol, 4.0 equiv.), **3a** (0.8 mmol, 2.0 equiv.), LiOMe (1.8 mmol, 4.5 equiv.) in solvent (2.5 mL), stirred at 60 °C, 10 h.

including -Cl (**4b**), -NMe₂ (**4c**), -CO₂Me (**4 d**), and -Bpin (**4e**), were well-tolerated, affording the corresponding diboraryl products in good to high yields. Notably, unprotected aniline (**4 f**) and alcohol (**4 g** and **4 h**) groups also underwent efficient reactions. Furthermore, different heteroaromatic compounds reacted effectively in this system, including unprotected indole (**4 m**), pyridine (**4n**), carbazole (**4o**), and thiazole (**4p**), with the target products obtained in moderate to excellent yields, demonstrating the broad functional group tolerance of this method. The relative configuration of product **4k** was confirmed by X-ray crystallography. We next examined the scope of vinyl triflates, all of which smoothly underwent the reaction. Specifically, nitrogen-containing cyclohexenyl triflate and 5-membered fused-ring vinyl triflate were successfully transformed into target products (**5b** and **5c**) with high efficiency. Addition-ally, sterically hindered bicyclic triflates and seven-membered ring substrates also showed good compatibility, affording the products (**5e, 5 f** and **5i**) in moderate yields, with the absolute configuration of product **5i** confirmed by X-ray analysis. Notably, 3- and 4-substituted vinyl triflates (**5 g** and **5 h**) successfully underwent the transformation, affording the target

products with excellent diastereoselectivity. The absolute configura-tions of the products were unambiguously confirmed by single-crystal X-ray diffraction. Remarkably, a structurally complex fused-ring vinyl triflate derived from santonin was also well tolerated, delivering the desired product (**5j**) in moderate yield. Furthermore, linear poly-substituted vinyl triflates (**5k** and **5 l**) smoothly participated in the reaction, further demonstrating the broad substrate scope and robustness of the methodology.

## Synthetic transformation and applications
The synthetic utility of this strategy was demonstrated as shown in Fig. 4. First, a gram-scale experiment was conducted, in which the aryl-diboration product **4a** was obtained in high yield (72%, 3.2 g) with excellent regio- and diastereoselectivity at a 10 mmol scale, high-lighting the scalability and practicality of the reaction. Second, several downstream transformations were explored (Fig. 4). For instance, compound **7**, featuring a [4.6]-spirocycle scaffold, was readily syn-thesized with excellent diastereoselectivity via a sequence of Zweifel olefination/oxidation/Heck reaction. Separately, *cis*-fused [6.5]- and

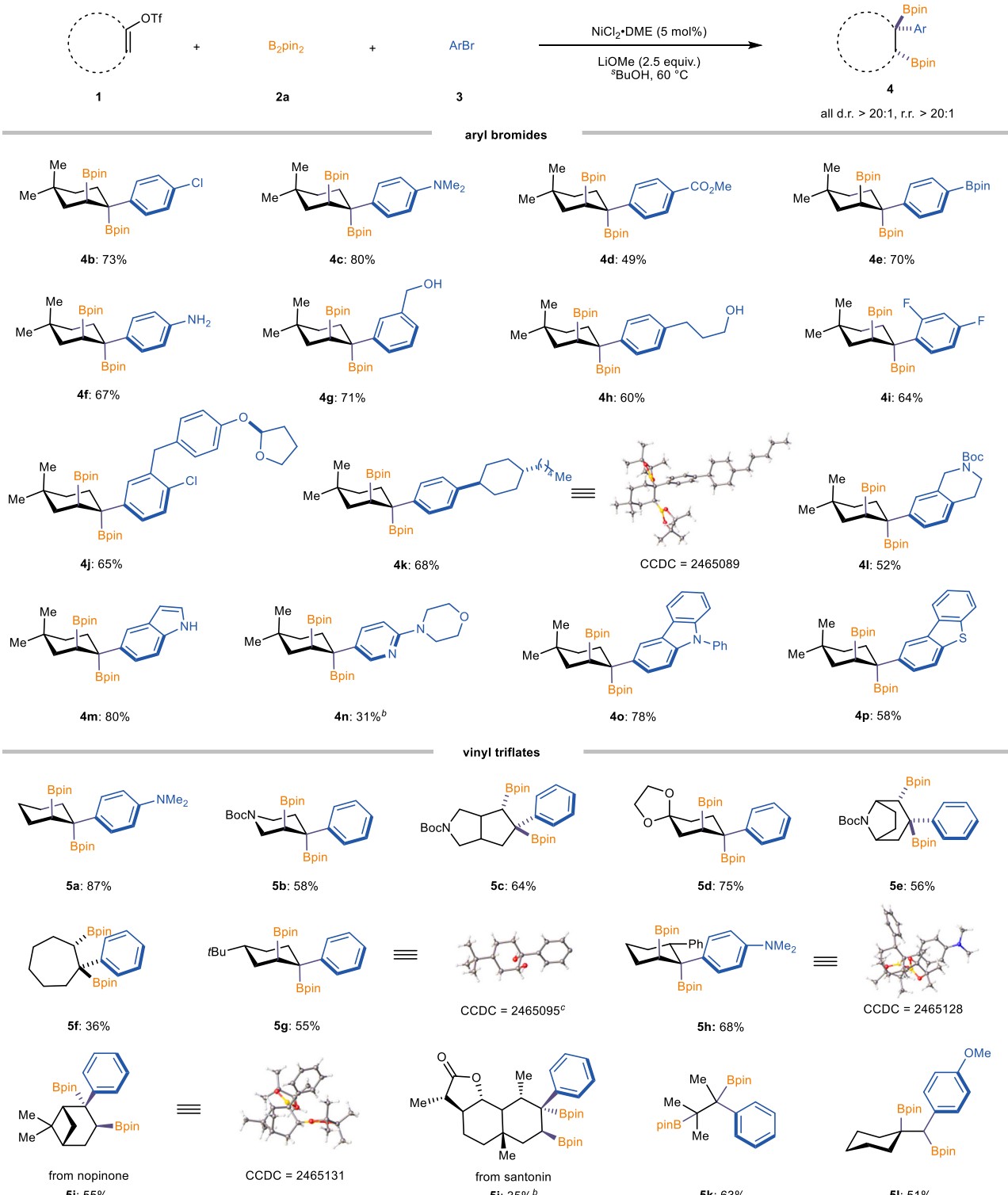

**Fig. 3 | Substrate scope.** [a]General reaction conditions: NiCl₂·DME (5.0 mol %), **1** (0.4 mmol, 1.0 equiv.), **2a** (1.6 mmol, 4.0 equiv.), **3** (0.8 mmol, 2.0 equiv.), LiOMe (1.8 mmol, 4.5 equiv.) in ˢBuOH (2.0 mL), stirred at 60 °C for 10–24 h. Given yield refers to the isolated yield of the single isomeric product. [b]Isolated yield of the corresponding alcohol after oxidation. [c]Single-crystal structure of the oxidation product.

[6.7]-bicyclic compounds **8** and **9** were synthesized in one pot through sequential boron oxidation and cyclization reactions, also with excellent diastereoselectivity.

Unsymmetrically ortho-aryl-substituted cyclohexanones are privileged motifs in numerous bioactive molecules, yet remain synthetically challenging to access[40–43]. Leveraging our strategy for molecular skeleton editing and late-stage functionalization, we conducted a series of arylative ketone displacement and migration reactions. As shown in Fig. 5, starting from simple ketones, the corresponding arylative ketone migration products **10**, **11**, and **13** were efficiently constructed in moderate yields via triflation, this nickel-catalyzed methodology, and subsequent oxidation/BF₃·Et₂O-mediated transformation. Notably, product

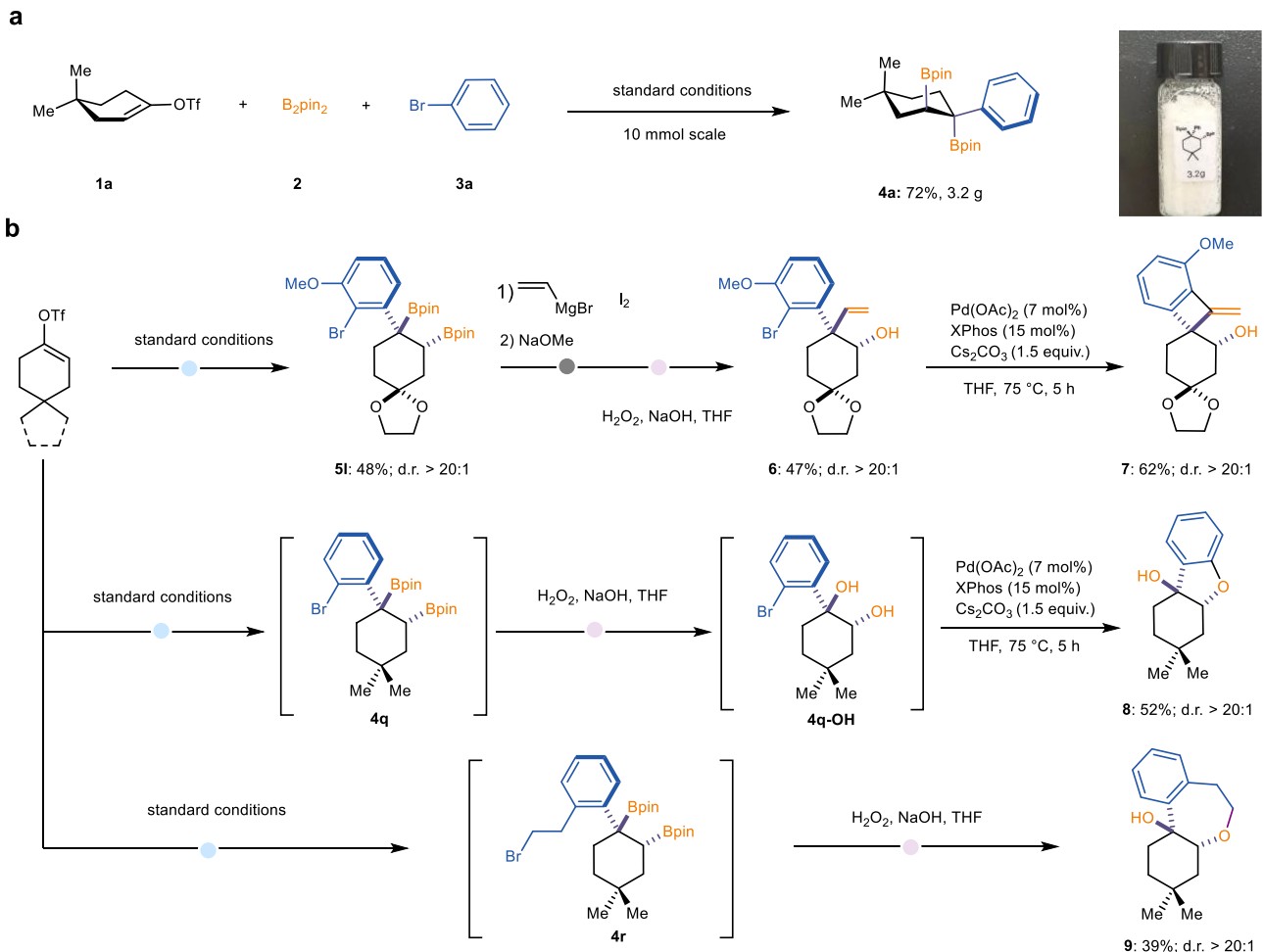

**Fig. 4 | Synthetic transformations. a** Scale-up reactions. **b** Diversification of representative products.

**11** was obtained with excellent *trans*-selectivity, and its structure was confirmed by single-crystal X-ray diffraction. Moreover, late-stage modification of the A-ring in steroids is a pivotal transformation in medicinal chemistry. Applying our protocol, we achieved the trifunctionalization of drostanolone propionate in moderate yield. Following oxidation and BF₃·Et₂O treatment, we isolated the arylative ketone migration product **15** with exceptional diastereoselectivity (>20:1 d.r.). Similarly, arylative ketone migration products **17** and **19** were synthesized in moderate yields and high stereoselectivity from dihydrocholesterol and epiandrosterone derivatives, respectively, via a one-pot, three-step sequence. To further demonstrate the synthetic utility of our platform, we designed a programmable, stepwise arylative ketone displacement and migration sequence. Starting from a simple cyclohexanone substrate **20**, the first arylation yielded 2-phenylcyclohexanone (**21**) in moderate yield. Subsequent iterative application of this reaction system successfully enabled the second, third, and fourth arylative ketone migration events, furnishing a series of highly functionalized cyclohexanone scaffolds bearing distinct aryl substituents (**22-24**), with excellent *trans*-diastereoselectivity observed at each step. Owing to pronounced steric hindrance, further installation of a fifth aryl group on the ring is not feasible under this protocol. Such structurally complex molecules are typically difficult to access using traditional synthetic methods, underscoring the broad applicability and strong synthetic potential of this strategy.

## Discussion

In summary, we have developed a nickel-catalyzed deoxygenative trifunctionalization of vinyl triflates that proceeds under mild conditions with excellent regio- and stereoselectivity. This strategy enables the efficient synthesis of a diverse set of boron-containing, polysubstituted cyclic compounds. Subsequent one-pot oxidation and BF₃·Et₂O-mediated rearrangement of these intermediates allows for programmable arylative ketone migration, substantially broadening the synthetic potential of this platform. This work offers a powerful approach for skeletal editing and the rapid diversification of molecular scaffolds, with direct relevance to the development of structurally and functionally diverse drug candidates. We anticipate that this methodology will find wide application in medicinal chemistry and related fields.

## Methods

### Representative procedure for trifunctionalization of vinyl triflates

Under a nitrogen atmosphere, an oven-dried 10 mL reaction tube equipped with a magnetic stir bar and sealed with a rubber stopper was used. Sequentially, NiCl₂·DME (4.4 mg, 0.02 mmol, 5 mol%), MeOLi (68.3 mg, 1.0 mmol, 4.5 equiv.) and B₂pin₂ (406.2 mg, 1.6 mmol, 4.0 equiv.). Then anhydrous ˢBuOH (1.5 mL), **1a** (0.4 mmol, 1.0 equiv.), **3a** (0.8 mmol, 2.0 equiv.) and anhydrous ˢBuOH (1.0 mL) were added in this order, and the mixture was stirred at 60 °C for 10−24 h. Then the mixture was filtered through a silica plug with ethyl acetate and concentrated. The crude material was separated on a silica gel column affording the desired product. For further details, please refer to the Supplementary Information.

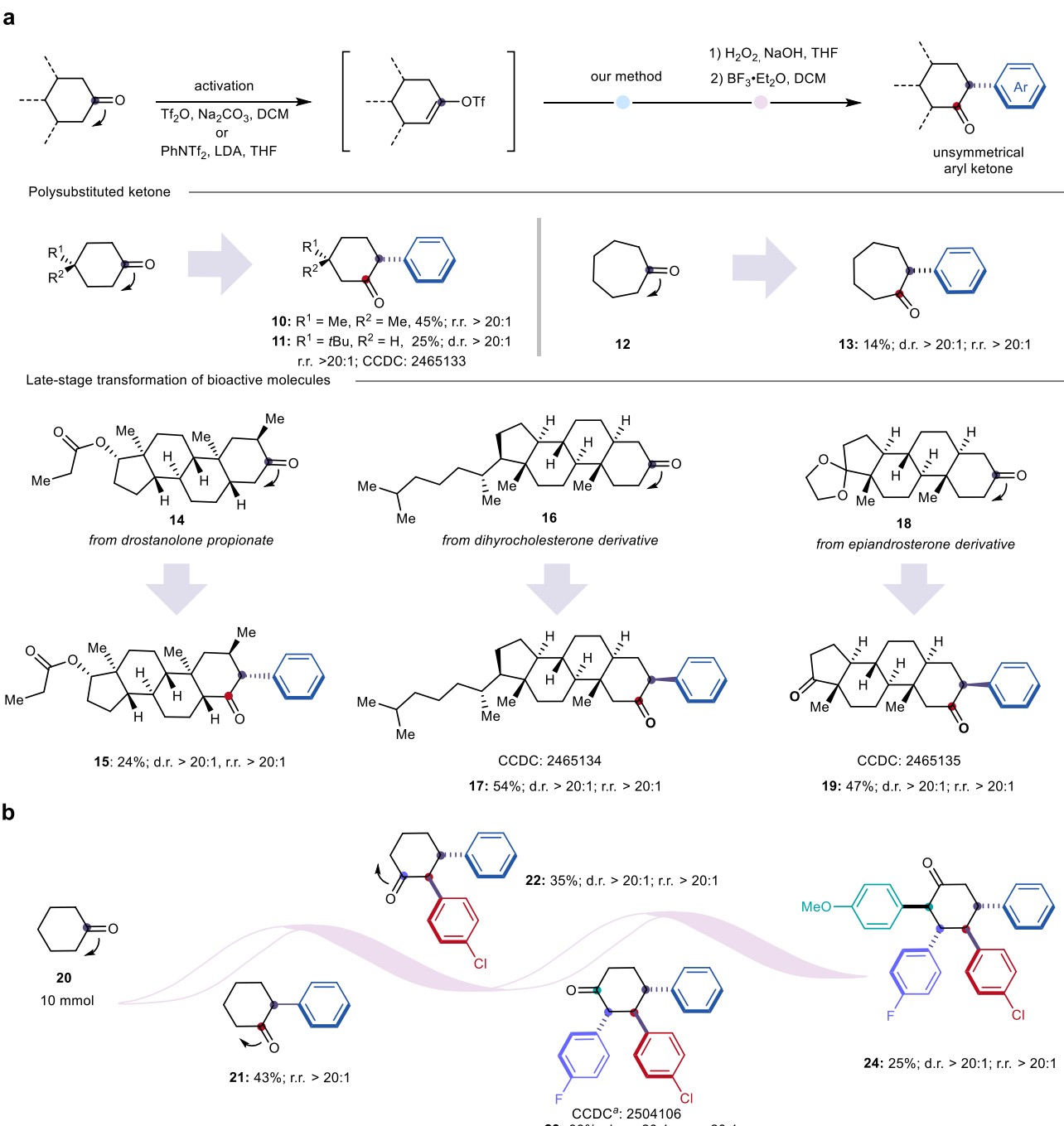

**Fig. 5 | Arylative ketone migration. a** Arylative ketone migration of polysubstituted ketone and bioactive molecules. **b** Programmable arylative ketone migration. $^a$Single-crystal structure of the trifunctionalization product.

## Data availability

The authors declare that all other data supporting the findings of this study are available within the article and Supplementary Information files and can also be obtained from the corresponding author on request. Crystallographic data for the structures reported in this Article have been deposited at the Cambridge Crystallographic Data Centre, under deposition numbers CCDC 2465089 (**4k**), CCDC 2465095 (**5g-OH**), CCDC 2465128 (**5 h**), CCDC 2465131 (**5i**), CCDC 2465133 (**11**), CCDC 2465134 (**17**), CCDC 2465135 (**19**) and CCDC 2504106 (**23-B**). Copies of the data can be obtained free of charge via https://www.ccdc.cam.ac.uk/structures/.

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

## Acknowledgements

National Natural Science Foundation of China (grant nos. 22122107 to G.Y., 22401220 to Y.-Y.L. 22371019 to Y.L.,) is acknowledged for financial support. Guangdong Basic and Applied Basic Research Foundation (grant nos. 2024A1515011689 to G.Y.). Fundamental Research Funds for the Central Universities (grant nos. 413100070 to Y.-Y.L.). We thank the Core Facility of Wuhan University for their assistance with X-ray crystallographic analysis.

## Author contributions

G.Y., Yangyang Li, and Yu Liu designed the project and directed the work; S.W. performed all synthetic experiments. T.Y. helped perform the experiments determing the substrates scope and synthetic applications.

All authors have contributed to the discussion of this work. G.Y., Yangyang Li, and S.W. wrote the manuscript.

## Competing interests

Authors declare no competing interests.
