## [Transparent Peer Review file · Nature Communications]

Ketone displacement and migration enabled by trifunctionalization of vinyl triflates

Corresponding Author: Professor Guoyin Yin

Version 0:

Reviewer comments:

Reviewer #1

(Remarks to the Author)

This manuscript describes a nickel-catalyzed deoxygenative trifunctionalization of vinyl triflates that enables the synthesis of boron-containing polysubstituted cyclohexanes featuring quaternary carbon centers under mild conditions. The methodology is robust, supported by extensive mechanistic studies, a broad substrate scope, and practical applications—including gram-scale synthesis and late-stage derivatization of complex bioactive molecules. Notably, formal arylative ketone migration is achieved via post-functionalization steps (oxidation/ $\text{BF}_3 \cdot \text{Et}_2\text{O}$ -mediated rearrangement), highlighting the method's utility in molecular skeleton editing. Particularly impressive is the demonstration of programmable arylative ketone migration (Fig. 5b), which underscores the versatility and synthetic value of this approach. In summary, I find this work highly compelling and support its publication in Nature Communications after minor revisions. Below are several suggestions to enhance clarity and impact:

1. The title and Introduction strongly emphasize functional group displacement and migration as the central theme. However, the core of the manuscript focuses on the development and scope of nickel-catalyzed trifunctionalization (Fig. 2-4). The ketone migration is a valuable yet secondary application that comes after further synthetic modification (Fig. 5). This disconnect may confuse readers. Additionally, Fig. 1c aims to illustrate the concept of ketone migration but does not explicitly show that the process requires additional 2 steps (oxidation and $\text{BF}_3 \cdot \text{Et}_2\text{O}$ treatment) following the initial trifunctionalization. Revising the title and Introduction to more accurately reflect the methodological advance, along with clarifying Fig. 1c to include the post-functionalization steps, would significantly improve conceptual clarity.

2. The structure of compound 7 in Figure 4b appears to be incorrectly drawn. The connectivity shown seems inconsistent with the described Heck reaction step. This should be carefully reviewed and corrected to avoid confusion.

3. The programmable arylative ketone migration demonstrated for compounds 20 to 23 is very impressive. I am curious whether any regioselectivity issues arise in the initial step of forming vinyl triflate from the α -arylated ketone.

Other comments:

The supporting information is in good shape.

Reviewer #2

(Remarks to the Author)

The authors describe a Ni-catalyzed borylation strategy for cyclic vinyl triflates to achieve carbonyl walking. This programmed arylative ketone migration proceeds with excellent diastereoselectivity and provides an iterative protocol for editing peripheral substituents on cyclic ketones. The manuscript is well-written, and the supporting data are robust. Thus, I recommend its publication in Nature Communications after minor revisions.

1. The specific reaction conditions for the downstream transformations and migrations in Figures 4 and 5 should be provided directly in the figure legends.

2. While the substrate scope for the initial borylation is abundant, the exploration of the ensuing arylative ketone migration is relatively limited. To better demonstrate the generality and utility of this method, the authors should expand this section to include various cyclic ketones, such as strained rings, fused rings, bridged rings, and macrocyclic systems in the Fig.5.

3. To further highlight the potential of this iterative method, is the synthesis of a sterically congested molecule like a hexa-aryl substituted cyclohexane feasible?

4. Please cite a relevant work on the programmed synthesis of tetra-aryl substituted alkenes via alkyne borylation strategy: "Nat. Commun. 2025, 16, 1025".

Reviewer #3

(Remarks to the Author)

This manuscript presents a transformative nickel-catalyzed protocol for the deoxygenative trifunctionalization of vinyl triflates, enabling efficient construction of boron-containing polysubstituted cyclohexanes featuring quaternary carbon centers. The work represents a paradigm shift in migratory functionalization chemistry, offering an exceptionally programmable approach with profound implications for complex molecule synthesis and medicinal chemistry applications.

Key Strengths and Innovations have been summarized here:

1. Methodological Breakthrough: First report of nickel-catalyzed arylboration coupled with ketone migration in vinyl triflates, elegantly solving the longstanding challenge of regioselective α -arylation in unsymmetrical ketones.
2. Unprecedented Stereocontrol: The "programmable diastereoselectivity" feature establishes new standards for stereochemical manipulation in polysubstituted cyclic systems.
3. Molecular Complexity Generation: Simultaneous installation of aryl, boron, and quaternary carbon centers provides exceptional scaffold diversification potential, directly addressing critical needs in medicinal chemistry ("scaffold hopping").
4. Technical Excellence: Comprehensive optimization studies and thorough characterization (including X-ray crystallography) strongly support the mechanistic proposal.

Overall, the manuscript is exceptionally well-prepared and merits publication in Nature Communications after addressing the following points:

1. The current focus on cyclohexanone derivatives is compelling, but demonstrating the protocol with other ring sizes (particularly cyclobutanone, cycloheptanone, and cyclooctanone derivatives) would significantly enhance the methodology's generality and impact.
2. For potential medicinal chemistry applications, it would be valuable to examine: The reactivity and selectivity of the cyclic vinyl triflate with β -substitution in acyclic system. Whether steric hindrance from β -substituents affects Ni-Bpin insertion efficiency.
3. Given the excellent diastereocontrol achieved, initial results with chiral ligands (even as proof-of-concept) would highlight the potential for enantioselective synthesis. How about the reactivity in nickel-catalyzed trifunctionalization of vinyl triflate using chiral ligand in this reaction?
4. The programmable nature of this transformation raises the important question of whether it could be extended to construct pentasubstituted cyclohexanones, which would represent a monumental advance in complex ring system synthesis.

Version 1:

Reviewer comments:

Reviewer #1

(Remarks to the Author)

The authors have addressed the reviewers' comments, and acceptance is recommended.

Reviewer #2

(Remarks to the Author)

Please add a note to the manuscript stating that the fifth aryl group cannot be installed on the ring via this protocol due to steric hindrance. This version is then approved for publication.

Reviewer #3

(Remarks to the Author)

The authors have solved the issues raised by reviewers. Therefore, I accept the current manuscript for publication in Nature Communications.

Response to Reviewers' Comments

Reviewer #1:

This manuscript describes a nickel-catalyzed deoxygenative trifunctionalization of vinyl triflates that enables the synthesis of boron-containing polysubstituted cyclohexanes featuring quaternary carbon centers under mild conditions. The methodology is robust, supported by extensive mechanistic studies, a broad substrate scope, and practical applications—including gram-scale synthesis and late-stage derivatization of complex bioactive molecules. Notably, formal arylative ketone migration is achieved via post-functionalization steps (oxidation/ $\text{BF}_3 \cdot \text{Et}_2\text{O}$ -mediated rearrangement), highlighting the method's utility in molecular skeleton editing. Particularly impressive is the demonstration of programmable arylative ketone migration (Fig. 5b), which underscores the versatility and synthetic value of this approach. In summary, I find this work highly compelling and support its publication in *Nature Communications* after minor revisions.

Our response: We greatly appreciate the reviewer for these positive comments and their agreement to have it published in *Nature Communications*.

1. The title and Introduction strongly emphasize functional group displacement and migration as the central theme. However, the core of the manuscript focuses on the development and scope of nickel-catalyzed trifunctionalization (Fig. 2-4). The ketone migration is a valuable yet secondary application that comes after further synthetic modification (Fig. 5). This disconnect may confuse readers. Additionally, Fig. 1c aims to illustrate the concept of ketone migration but does not explicitly show that the process requires additional 2 steps (oxidation and $\text{BF}_3 \cdot \text{Et}_2\text{O}$ treatment) following the initial trifunctionalization. Revising the title and Introduction to more accurately reflect the methodological advance, along with clarifying Fig. 1c to include the post-functionalization steps, would significantly improve conceptual clarity.

Our Response: We thank the reviewer for the valuable suggestions. In response, we have revised Fig. 1c to include the two key steps that occur after the trifunctionalization (oxidation and $\text{BF}_3 \cdot \text{Et}_2\text{O}$ treatment) and to more clearly illustrate how vinyl triflates, conveniently prepared from ketone substrates, evolve through subsequent functionalization. Regarding the reviewer's concern about a potential disconnect between the nickel-catalyzed trifunctionalization and the ensuing ketone migration, we would like to further clarify that the trifunctionalized intermediates serve not only as important precursors for constructing highly substituted cyclic structures that are difficult to access by conventional methods, but also as key platforms enabling the ketone displacement and migration sequence. Given the potential of this new method to enhance molecular complexity, we chose to present the overall workflow. The ketone-migration transformation is specifically highlighted because it represents a previously unreported synthetic strategy—one that fills a gap in existing methodology and enables asymmetric α -arylation of ketones with excellent regio- and stereocontrol. We further expanded the substrate scope for the ketone-migration study to showcase the unique advantages of this new transformation in constructing complex architectures, rather than repeating routine examples already covered in the trifunctionalization scope. We believe that these clarifications make the logical connection and complementarity between the two parts much clearer. We

believe these modifications greatly enhance the conceptual coherence and readability of the manuscript, and we thank the reviewer again for helping us improve the clarity of our work.

2. The structure of compound **7** in Figure 4b appears to be incorrectly drawn. The connectivity shown seems inconsistent with the described Heck reaction step. This should be carefully reviewed and corrected to avoid confusion.

Our Response: We sincerely thank the reviewer for this valuable suggestion. We have corrected the structure of compound **7** in Fig. 4b accordingly and have carefully reviewed the entire manuscript.

3. The programmable arylative ketone migration demonstrated for compounds **20** to **23** is very impressive. I am curious whether any regioselectivity issues arise in the initial step of forming vinyl triflate from the α -arylated ketone.

Our Response: In response to the reviewer's question, we clarify that the α -arylated ketone was alkylated using the bulky base LDA, resulting in a highly selective transformation. No regioisomeric products were observed experimentally.

Reviewer #2:

The authors describe a Ni-catalyzed borylation strategy for cyclic vinyl triflates to achieve carbonyl walking. This programmed arylative ketone migration proceeds with excellent diastereoselectivity and provides an iterative protocol for editing peripheral substituents on cyclic ketones. The manuscript is well-written, and the supporting data are robust. Thus, I recommend its publication in *Nature Communications* after minor revisions.

Our response: We greatly appreciate the reviewer for these positive comments and their agreement to have it published in *Nature Communications*.

1. The specific reaction conditions for the downstream transformations and migrations in Figures 4 and 5 should be provided directly in the figure legends.

Our Response: We thank the reviewer for this valuable suggestion. The captions of Figures 4 and 5 have been updated to include the specific reaction conditions for the downstream transformations and migrations.

2. While the substrate scope for the initial borylation is abundant, the exploration of the ensuing arylative ketone migration is relatively limited. To better demonstrate the generality and utility of this method, the authors should expand this section to include various cyclic ketones, such as strained rings, fused rings, bridged rings, and macrocyclic systems in the Fig.5.

Our Response: We sincerely thank the reviewer for their valuable suggestion. Following the reviewer's advice, we examined several representative cyclic ketone substrates, including strained, fused, bridged, and medium-sized rings, to further evaluate the generality of the arylative ketone migration reaction. However, the reactivity was found to be strongly dependent on the ring structure. For strained-ring substrates, the yield of the corresponding trifunctionalized intermediates was very low, and the arylative ketone migration products could not be isolated. In the case of bridged-ring substrates, aromatization occurred during the

rearrangement process, resulting in the loss of the desired migration products. For fused-ring substrates, ring-contraction products were observed instead of the expected migration products. In contrast, the seven-membered ring substrate successfully afforded the desired migration product, while the eight-membered ring substrate did not undergo the reaction under the current conditions. These results have been added to the revised manuscript and Supplementary Information (please see Fig. 5 & Supplementary Information).

3. To further highlight the potential of this iterative method, is the synthesis of a sterically congested molecule like a hexa-aryl substituted cyclohexane feasible?

Our Response: We thank the reviewer for this insightful suggestion. We attempted to further extend the iterative method to the construction of more highly substituted cyclohexanes. Using this strategy, we successfully synthesized tetra-aryl substituted cyclohexanes. However, when we attempted to construct penta-aryl substituted cyclohexanes, the significantly increased steric hindrance prevented the formation of the trifunctionalized intermediate required for the key ketone-migration step, and the reaction did not proceed. These results have been added to the revised manuscript and Supplementary Information (please see Fig. 5 & Supplementary Information).

b. Programmable arylative ketone migration

4. Please cite a relevant work on the programmed synthesis of tetra-aryl substituted alkenes via alkyne borylation strategy: "Nat. Commun. 2025, 16, 1025".

Our Response: We sincerely thank the reviewer for this suggestion. The manuscript has been updated to include a citation of Nat. Commun. 2025, 16, 1025.

Reviewer #3:

This manuscript presents a transformative nickel-catalyzed protocol for the deoxygenative trifunctionalization of vinyl triflates, enabling efficient construction of boron-containing polysubstituted cyclohexanes featuring quaternary carbon centers. The work represents a paradigm shift in migratory functionalization chemistry, offering an exceptionally programmable approach with profound implications for complex molecule synthesis and medicinal chemistry applications. Key Strengths and Innovations have been summarized here:

1. Methodological Breakthrough: First report of nickel-catalyzed arylboration coupled with ketone migration in vinyl triflates, elegantly solving the longstanding challenge of regioselective α -arylation in unsymmetrical ketones.

2. Unprecedented Stereocontrol: The "programmable diastereoselectivity" feature establishes new standards for stereochemical manipulation in polysubstituted cyclic systems.

3. Molecular Complexity Generation: Simultaneous installation of aryl, boron, and quaternary carbon centers provides exceptional scaffold diversification potential, directly addressing critical needs in medicinal chemistry ("scaffold hopping").

4. Technical Excellence: Comprehensive optimization studies and thorough characterization (including X-ray crystallography) strongly support the mechanistic proposal.

Overall, the manuscript is exceptionally well-prepared and merits publication in Nature Communications after addressing the following points:

Our response: We sincerely thank the reviewer for the positive evaluation of our work and for recognizing the quality of our manuscript. We appreciate the reviewer's thoughtful comments and constructive suggestions. We have carefully addressed all the points raised, as detailed below. We believe these revisions have further improved the clarity and scientific rigor of the manuscript.

1. The current focus on cyclohexanone derivatives is compelling, but demonstrating the protocol with other ring sizes (particularly cyclobutanone, cycloheptanone, and cyclooctanone derivatives) would significantly enhance the methodology's generality and impact.

Our Response: We sincerely thank the reviewer for this insightful suggestion. Following the reviewer's advice, we further examined substrates derived from cyclobutanone, cycloheptanone, and cyclooctanone to evaluate the generality of this transformation. Due to the steric hindrance from the substituents on cyclobutanone, the reaction proceeded with very low efficiency, and the desired product could not be obtained. The reaction with cycloheptanone derivatives proceeded smoothly, yielding the expected product. However, the cyclooctanone substrate did not undergo the transformation under the current conditions. These results have

been added to the revised manuscript and Supplementary Information (please see Fig. 3 & Supplementary Information).

2. For potential medicinal chemistry applications, it would be valuable to examine: The reactivity and selectivity of the cyclic vinyl triflate with β -substitution in acyclic system. Whether steric hindrance from β -substituents affects Ni-Bpin insertion efficiency.

Our Response: We appreciate the reviewer's valuable suggestion. In the revised manuscript, we further expanded our investigation to the acyclic substrates **5k** and **5l** (Figure 3) to evaluate the influence of β -substituents on the reaction activity and regioselectivity. The results indicate that even with increasing steric hindrance at the β -position, the efficiency of the Ni-Bpin insertion remains unaffected.

3. Given the excellent diastereocontrol achieved, initial results with chiral ligands (even as proof-of-concept) would highlight the potential for enantioselective synthesis. How about the reactivity in nickel-catalyzed trifunctionalization of vinyl triflate using chiral ligand in this reaction?

Our Response: We sincerely appreciate the reviewer's valuable suggestion. Preliminary attempts using several commonly employed chiral ligands resulted in significant suppression of the reaction reactivity. Further investigation on the enantioselective version of this transformation will be pursued in our future studies.

4. The programmable nature of this transformation raises the important question of whether it could be extended to construct pentasubstituted cyclohexanones, which would represent a monumental advance in complex ring system synthesis.

Our Response: We sincerely thank the reviewer for this insightful suggestion. We explored the extension of our iterative method toward the synthesis of more highly substituted cyclohexanes. Using this approach, tetra-aryl substituted cyclohexanes were successfully constructed. However, attempts to synthesize penta-aryl substituted cyclohexanes were unsuccessful, as the substantially increased steric hindrance hindered the formation of the trifunctionalized intermediate necessary for the key ketone-migration step, preventing the reaction from proceeding. These findings have been incorporated into the revised manuscript and Supplementary Information (see Fig. 5 and Supplementary Information).

b. Programmable arylative ketone migration

we wholeheartedly appreciate the reviewer's constructive comments on this work, and we would like to express our sincere gratitude for providing us with the opportunity to enhance our paper.

In conclusion, we sincerely thank the reviewer for recognizing our work and for providing valuable comments, which have greatly helped us further improve this research. We hope that our responses satisfactorily address all of your concerns.

Sincerely,

Guoyin Yin

Response to Reviewers' Comments

Reviewer #1 (Remarks to the Author):

The authors have addressed the reviewers' comments, and acceptance is recommended.

Our Response: We sincerely thank you for your thorough review of our revised manuscript and for your positive feedback. Your valuable comments were crucial in helping us improve our work, and we greatly appreciate your recommendation to publish our research in Nature Communications.

Reviewer #2 (Remarks to the Author):

Please add a note to the manuscript stating that the fifth aryl group cannot be installed on the ring via this protocol due to steric hindrance. This version is then approved for publication.

Our Response: We thank the reviewer for the valuable suggestion. We have added a note in the manuscript clarifying that, due to pronounced steric hindrance, the installation of a fifth aryl group on the ring is not feasible under the present reaction conditions.

Reviewer #3 (Remarks to the Author):

The authors have solved the issues raised by reviewers. Therefore, I accept the current manuscript for publication in Nature Communications.

Our Response: We deeply appreciate the reviewer for his/her constructive suggestions and positive comments to improved enhancement of our research. We are very pleased that our reply has satisfied the reviewers.